# The Gut Microbiome and Alcoholic Liver Disease: Ethanol Consumption Drives Consistent and Reproducible Alteration in Gut Microbiota in Mice

**DOI:** 10.3390/life11010007

**Published:** 2020-12-24

**Authors:** Erick S. LeBrun, Meghali Nighot, Viszwapriya Dharmaprakash, Anand Kumar, Chien-Chi Lo, Patrick S. G. Chain, Thomas Y. Ma

**Affiliations:** 1Biosecurity and Public Health, Los Alamos National Laboratory, Los Alamos, NM 87545, USA; elebrun@lanl.gov (E.S.L.); akumar@lanl.gov (A.K.); chienchi@lanl.gov (C.-C.L.); 2Department of Medicine, Division of Gastroenterology and Hepatology, Penn State College of Medicine, Hershey, PA 17033, USA; mnighot@pennstatehealth.psu.edu (M.N.); vdharmaprakash@pennstatehealth.psu.edu (V.D.); 3Department of Internal Medicine, University of New Mexico School of Medicine, Albuquerque, NM 87131, USA

**Keywords:** gut microbiome, microbial ecology, indicator species, bacterial communities, mouse model, alcohol effects, Alcoholic Liver Disease, microbiome, ethanol-induced liver disease, leaky gut

## Abstract

Phenotypic health effects, both positive and negative, have been well studied in association with the consumption of alcohol in humans as well as several other mammals including mice. Many studies have also associated these same health effects and phenotypes to specific members of gut microbiome communities. Here we utilized a chronic plus binge ethanol feed model (Gao-binge model) to explore microbiome community changes across three independent experiments performed in mice. We found significant and reproducible differences in microbiome community assemblies between ethanol-treated mice and control mice on the same diet absent of ethanol. We also identified significant differences in gut microbiota occurring temporally with ethanol treatment. Peak shift in communities was observed 4 days after the start of daily alcohol consumption. We quantitatively identified many of the bacterial genera indicative of these ethanol-induced shifts including 20 significant genera when comparing ethanol treatments with controls and 14 significant genera based on temporal investigation. Including overlap of treatment with temporal shifts, we identified 25 specific genera of interest in ethanol treatment microbiome shifts. Shifts coincide with observed presentation of fatty deposits in the liver tissue, i.e., Alcoholic Liver Disease-associated phenotype. The evidence presented herein, derived from three independent experiments, points to the existence of a common, reproducible, and characterizable “mouse ethanol gut microbiome”.

## 1. Introduction

Alcoholic Liver Disease (ALD) remains one of the most important diseases in the US. Overall, ALD affects 5 to 7 million US adults of which approximately 1.4 million adults receive treatment and close to 88,000 people die each year of alcohol-related causes [1]. Due to a lack of effective therapies for treatment [2], ALD remains a major health problem worldwide. Recent scientific studies have suggested that ethanol-associated alterations in the gut microbiome are an important pathogenic factor contributing to the development of ALD. The mechanistic causes and effects of gut dysbiosis remain an important area of investigation [3]. Numerous human and animal studies have shown that alcohol affects the gut–liver axis at multiple interconnected levels, including alterations in the gut microbiome and decreases in intestinal epithelial barrier function leading to an increase in systemic exposure of gut bacteria and bacterial toxin culminating in the development of ALD [4,5,6].

In human subjects exhibiting alcohol-related gut microbe dysbiosis, the observed increase in specific bacterial populations has been associated with gut leakiness. Intestinal dysbiosis in heavy alcohol drinkers involves, in particular, the enrichment of Enterobacteriaceae and decline of Bacteroidetes and Lactobacillus [7]. Similarly, mouse models of ALD have also demonstrated a decline in the abundance of both Bacteroidetes and Firmicutes along with a proportional increase in Gram-negative Proteobacteria and Gram-positive Actinobacteria [8]. A number of specific genera level taxa have been correlated to both alcohol-related microbiome shifts and ALD phenotypes. Interestingly, endogenous populations of documented ALD preventative bacterial genera such as Lactobacillus and Bifidobacterium have also been shown to be depleted in alcohol-fed mice models [7].

The important role of gut microbiota in ALD pathogenesis suggests the need for continued development of novel and innovative approaches to help explain the disparate observations across various animal model systems and human studies. A comprehensive understanding of how the microbial community affects ALD development is necessary to develop therapeutic strategies to target the gut microbiome in order to treat ALD.

Here we look at gut microbiome shifts in a mouse model of ALD, consisting of chronic ethanol feeding plus binge (Gao-binge model) across three independent experiments, in order to explore treatment-based and temporal effects related to alcohol consumption and ALD. This study design allows us to account for potential spurious experiment-related microbiome differences and focus on microbiome changes associated specifically with ethanol treatment and related temporal shifts.

## 2. Methods

### 2.1. Experimental Design

Eight to 12-week-old male C57BL/6N mice were divided into two groups (control and ethanol treatment) and were fed a nutritionally adequate liquid control diet (Lieber DiCarli Control Diet, Bioserv, Frenchtown, NJ, USA) [9] for 3 consecutive days to establish a similar baseline gut microbiome. Fecal samples were collected during this acclimation period (day 0). This was followed by feeding with Lieber DiCarli Diet containing 5% ethanol for 10 days in ethanol group mice while control mice were pair-fed with Lieber DiCarli diet without ethanol for the same 10 days (days 1–10). Mice were kept in individual cages with individual feeders and three independent experiments named ALD1, ALD2, and ALD3 were performed at different times. A total of 26 mice (13 control and 13 ethanol-treated) were included in this study. ALD1 consisted of 6 mice (3 control and 3 ethanol-treated) while ALD2 and ALD3 each consisted of 10 mice (5 control and 5 ethanol-treated). In each experiment, the age of the mice and experimental conditions were the same. The control and ethanol diets were prepared fresh each day and delivered in individual, autoclaved feeders. Fecal samples were collected from individual mice cages each day and stored at −20 °C for DNA extraction. A new, clean cage was interchanged for the following day’s fecal sample collection. On the 11th day of ethanol feeding, mice in the ethanol group received a gavage of a single dose of ethanol (5 g/kg body weight), while mice in the control group received a gavage of isocaloric dextrin maltose. After the 11th day, mice were sacrificed for liver histopathology. Due to insufficient fecal material for DNA extraction after day 11 gavage, related samples were not analyzed for associated microbiome changes. A schematic summary of the study can be viewed in Appendix A.

### 2.2. Ethics

Mouse studies were approved by the Pennsylvania State University, College of Medicine, Hershey, PA, USA, Institutional Animal Care and Use Committee (IACUC protocol-Role of intestinal permeability in Alcoholic Liver disease-ALD-PROTO201900704).

### 2.3. DNA Extraction and Sequencing

Fecal DNA was isolated using the Qiagen pro DNA kit according to the manufacturer’s instructions modified to increase bead-beating time to 15 min. 16S amplicon DNA was sequenced at Los Alamos National Laboratory (Los Alamos, NM, USA). Degenerate primers amplifying the V3-V4 region of bacterial rRNA genes were used (F341-806R pair). PCR amplification was performed in two rounds, the first to create an amplicon library using KAPA HiFi HotStart Ready Mix (KAPA Biosystems, Inc., Cape Town, South Africa) and the second to add Nextera XT v2 indexes (Illumina, San Diego, CA, USA) as described in previous work [8]. Detailed PCR information is available in the Appendix A. Clean-up and quality control were performed on amplicon libraries as detailed in the Appendix A. Sequencing was performed on an Illumina MiSeq platform sequencer.

### 2.4. Sequence Data Processing

QIIME2 version 2019.10 was applied to demultiplexed sequencing data using the DADA2 plug-in to denoise the data [10,11]. The pipeline included quality trimming, denoising, read joining, amplicon sequence variant (ASV) determination, chimera removal, and taxonomic classification of ASVs using a pre-trained Greengenes 13_8 99% 16S rRNA Naive Bayes classifier [12]. Further details of pipeline parameters and steps are available in the Appendix A. Samples with fewer than 1000 reads in rarefaction were discarded. Operational taxonomic units (OTUs) were constructed from ASVs classified to the same genera taxonomic level.

### 2.5. Data Analysis

Data analysis was performed using R data analysis software version 4.0.2 [13]. Figures were generated using the ggplot2 package version 3.3.2 in R [14]. MANOVA-type analysis was performed using the vegan package version 2.5.6 in R [15]. All MANOVA-type analysis was performed using relative abundance and 1000 permutations. Non-metric multidimensional scaling (NMDS) was performed on relative abundance data with Bray–Curtis distances using the vegan package in R. Temporal generalized additive models (GAMs) were built using ordisurf from the vegan package in R. Diversity (Shannon and Simpson indices) and beta dispersion (Bray–Curtis distance) testing (Kruskal–Wallis and Tukey HSD respectively) were performed using the vegan package in R. Point-biserial coefficient analysis (PBCA) was performed on treatments and temporal data using the indicspecies package version 1.7.9 in R [16]. All PBCA was performed using relative abundance with 1000 permutations. Heatmap figures for temporal PBCA and mean abundance differences were made using the pheatmap package version 1.0.12 in R [17]. Peak shift temporally in ethanol treatments was assessed with a bootstrap value of 500 using the TITAN2 package version 2.4 in R [18]. TITAN2 was designed to identify environmental thresholds by using indicator species scores to integrate occurrence, abundance, and directionality of taxa responses [18]. Here, we have adapted TITAN2 to a temporal exploration by using days post-treatment or “time” as our ecological gradient. Cluster analysis was performed using the complete linkage method of hclust through vegan on a Bray–Curtis distance matrix and cutting the tree into a priori best fit groups of *k* groupings all within the vegan package in R. *k* cluster ranges were tested using PARMANOVA and ANOSIM via vegan in R in order to maximize significant test statistics and find optimal *k*.

## 3. Results

### 3.1. S Amplicon Sequencing Summary

16S amplicon sequencing and processing of 286 samples representing days 0 through 10 of treatment across three independent experiments (ALD1 = 66, ALD2 = 110, ALD3 = 110) resulted in 8,806,863 paired end Illumina reads of which 5,646,371 passed strict filtering and quality control methods for an average of 19,743 reads per sample. Two samples, including one day 9 control from ALD3 and one ethanol treatment sample for day 8 from ALD1 were removed due to having fewer than 1000 rarified reads. The 26 samples from day 0 were analyzed to test starting community similarity. Fecal samples from day 0 showed no significant difference in PERMANOVA or ANOSIM tests between control and ethanol groups. 258 samples were used for treatment day 1–10 from the experiments (ALD1 = 59, ALD2 = 100, ALD3 = 99) for community analysis. In total, 1145 unique bacterial amplicon sequence variants (ASVs) were classified across these samples.

### 3.2. Ethanol Treatment vs. Controls

Ethanol group bacterial communities and those of controls were significantly different from each other using both PERMANOVA (F = 20.2, *p* < 0.001) and ANOSIM (R = 0.17, *p* < 0.001) tests. Differences between experimental groups and between experiments were visualized in NMDS ordination (Figure 1). Across experiments, a significant decrease in mean Shannon (χ^2^ = 8.3, *p* = 0.004) and Simpson (χ^2^ = 18.9, *p* < 0.001) alpha diversity scores were observed for samples in the ethanol group (Appendix A). We also observed a significant decrease in beta dispersion (diff = −0.02, *p* = 0.02) in the ethanol group across the 3 experiments (Appendix A). PBCA identified 315 significant ASVs (control = 150, ethanol = 165) which was representative of approximately a third of total ASVs. PBCA identified 20 significant genera OTUs (control = 11, ethanol = 9) that are shown in Table 1 and include genera shown in Figure 2 as well as in detailed mean relative abundance plots (Appendix A).

### 3.3. Temporal Community Shifts

A temporal GAM model on the NMDS ordination significantly explains 16.1% of deviance (Adj, r^2^ = 0.15, *p* < 0.001) and is shown in Figure 1. However, while a temporal GAM model on only control samples explains negligible deviance (<0.001%), when applied to only ethanol samples, the temporal model explains 40.2% of deviance (Adj. r^2^ = 0.37, *p* < 0.001). PBCA on controls temporally identified 8 significant genera OTUs (Figure 3). PBCA on ethanol treatments identified 6 significant genera OTUs (Figure 3). The strength of abundance difference for these genera is shown in Figure 2 and detailed mean relative abundance plots are shown in Appendix A. TITAN2 analysis shows peak community shift occurring around day 4 (Appendix A). Importantly, the 8 significant taxa identified by TITAN2 fully overlap those found by PBCA (Appendix A).

### 3.4. Histopathology of Liver

The alcohol feed/binge caused macrovascular steatosis of hepatocytes and inflammatory infiltration consistent with alcoholic steatohepatitis in the ethanol group compared to controls by day 11 of the study (Figure 4).

### 3.5. Cluster Analysis

We tested a range of 2 to 10 *k* clusters for optimal a priori clustering. Three, 4, and 5 groups were selected as the optimal group numbers in the range based on PERMANOVA and ANOSIM statistics (Appendix A). *k* = 3 groupings was observed to be representative of the 3 experiments ALD1, ALD2, and ALD3 (Appendix A). *k* = 4 groupings demonstrated ALD1 ethanol treatments emerging as the 4th group (Appendix A). *k* = 5 groupings demonstrated ALD2 ethanol treatments (with some control samples) emerging as the 5th grouping (Appendix A).

## 4. Discussion

This study allows us to identify three of the primary factors affecting community composition/assemblage in our data sets, namely experiment, treatment, and time. From our cluster analysis, NMDS visualization, and significant but lower ANOSIM statistic in comparing treatments, we find that communities in experiments are significantly different and variable despite the near identical treatment regimes. By combining samples from the three experiments and focusing on treatment and temporal effects, we increase our confidence in our findings being specific to ethanol treatment and not due to any unique experimental artifact, while also reducing our potential false positive rate.

In addition to a significant difference in community structure between control and ethanol group communities, we also see a decrease in community alpha and beta diversities. While per-individual microbial diversity decrease has previously been reported, other studies have actually shown an increase in alpha diversity [19,20,21,22,23,24]. It may well be that a shift is expected to happen but that the direction of the shift is dependent on the pre-treatment community or experimental design effects. Previous studies have pointed to microbiome colonization changes induced by ethanol treatment [25]. Decreases in alpha diversity have generally been associated with dysbiosis and disease phenotypes in both humans and mice.

Beta diversity changes have also been well documented [19,20,21,22,23,24]. A decrease in beta dispersion in our study, paired with the temporal GAM model explaining deviance primarily in ethanol-treated samples suggests that ethanol-treated microbiomes are converging towards a more similar microbiome, even across multiple experiments. It is possible that this effect is due to impacts on microbial colonization with the mice in our three independent experiments being fed the same diet under the same conditions. Increases seen in beta dispersion in previous studies may be artifactual of conditions within any single experiment. It will be prudent to continue to explore ethanol-induced diversity shifts across a large number of studies and experiments to explore if all ethanol-induced microbial communities in mice shift towards some common taxonomic composition. Important taxa certainly appear largely conserved across studies as outlined below.

The current body of literature includes studies that both account for and ignore temporal factors [19,20,21,22,23,24,25,26]. Our study suggests that temporal factors are critical to understanding the effects of ethanol on gut microbiota. Temporal factors appear unique to ethanol consumption compared with controls and a focus only on treatment-based changes may ignore important temporal dynamics of gut microbiota shifts. Perhaps surprisingly, the initial impact of ethanol temporally on significant taxa appears to be on Gram-positive bacteria with some recovering and thriving while others do not recover.

In our study, we observe a fair amount of overlap in PBCA significant genera between the treatment-based analysis and the temporal analysis. However, the information provided by each analysis is unique and does not tell us the same thing about the genera in question. For example, *Clostridium* appears as a significant organism for the ethanol group as a whole. This suggests that there is significantly more relative *Clostridium* in all ethanol-treated mice. It also appears as a significant genus in both control and ethanol temporal analyses. However, from Figure 3 and Appendix A, we see that this shift is similar in both groups meaning that while *Clostridium* is likely important in the context of ethanol consumption, it is not relevant temporally. Closer investigation of significant genera allows us to bin identified genera into three loose classifications. These include genera that are seen in greater relative abundance primarily in controls vs. primarily in ethanol groups, genera that shift significantly in ethanol treatments but not in controls, and genera that shift significantly in both control and in ethanol groups but shift antithetically.

Genera that are observed in significantly higher relative abundance in all control or ethanol samples are presented in Table 1. The aforementioned *Clostridium* is an example of a genus found in significantly higher relative abundance in all ethanol-treated mice (Figure 2). Significant genera in controls are fairly evenly distributed between Gram-positive and Gram-negative bacteria (G+ = 5, G− = 6). Significant genera in ethanol treatments favor Gram-negative bacteria (G+ = 3, G− = 6). It is important to note that of the three significant Gram-positive genera for ethanol treatments, *Adlercreutzia* (Table 1, Figure 2 and Figure 3, Appendix A) and *Enterococcus* (Table 1, Figure 2 and Appendix A) did not establish larger relative abundance populations until later in the treatment time course (~day 4 and ~day 5 respectively), again highlighting the importance of including temporal considerations. *Adlercreutzia* has been associated with fatty liver disease and obesity in mice [27,28]. Higher levels of ethanol group Gram-negative bacteria (*Prevotella*, *Bilophila*, *Desulfovibrio*, and *Helicobacter*) have been associated with deleterious health phenotypes such as liver disease due to the known liver inflammation promoting effects of LPS [29,30,31,32].

Two Gram-positive genera, *Lactobacillus* (Figure 2 and Figure 3, Appendix A) and *Turicibacter* (Figure 2 and Figure 3, Appendix A) shift significantly in the ethanol group but not in controls. Both were observed in significant relative abundance on day 1, the day of ethanol introduction, and then steeply decline, remaining negligible in relative abundance across the remainder of the experimental time course. *Lactobacillus* has previously been studied in the context of alcohol disease primarily as a mitigative probiotic or prophylactic protective of liver disease phenotypes [33,34,35], consistent with our findings. Decreased *Turicibacter* abundance has been associated with anxiety-like effects and social avoidance behavior in mice as well as reduced intestinal cytokine expression and obesity and fatty liver disease [36,37,38,39].

Three Gram-positive bacteria, *Adlercreutzia* (Table 1, Figure 2 and Figure 3, Appendix A), *Allobaculum* (Figure 2 and Figure 3, Appendix A), and *Bifidobacterium* (Figure 2 and Figure 3, Appendix A) were found to shift significantly in both controls and ethanol treatments but shift antithetically temporally. *Adlercreutzia* relative abundance is similar in control and ethanol groups days 4 through 6 but then declines in the controls yet increases in ethanol treatments. *Allobaculum* is abundant in controls during the first half of the time course but then declines to negligible levels for the second half of the experiment while being negligible in ethanol treatments for the first half of the time course and growing to significant levels for the second half. *Bifidobacterium* abundance behaves similarly to *Allobacullum*. These bacteria were possibly knocked out by initial treatment and then not only recovered but thrived over the time course. *Allobaculum*, similar to *Adlercreutzia*, has been previously associated with fatty liver disease phenotypes [36,40,41]. Similar to *Lactobacillus*, *Bifidobacterium* has primarily been explored as a probiotic to mitigate adiposity and inflammation associated with fatty liver disease [42,43].

*Oscillospira* and *Dorea* were only seen as significant genera in temporal shifts in controls and trends in their relative abundance plots are difficult to discern (Appendix A respectively). *Dorea* has been identified as significant in other studies [44]. The story of these two genera is difficult to further define based on our data.

Despite not being a significant genus in our analysis, we also took a closer look at *Akkermansia* as a genus of interest due to its prevalence in the literature with some studies showing an ethanol-related increase and others showing a depletion [45,46,47]. In our study, we do see an increase to a peak in relative abundance in ethanol treatments on day 3 which then declined (Figure 3, Appendix A). This temporal variability in *Akkermansia* abundance may explain why some studies have reported increases in abundance while others have reported decreases in abundance associated with ethanol treatment.

Despite the use of different methods and approaches, our findings overlap with bacterial genera found in other recent studies of ethanol-treated mice. Bluemel et al. identified *Bacteroides*, *Prevotella*, *Parabacteroides*, *Blautia*, and *Lactobacillus* as displaying significant changes in relative abundance [25]. While these genera are also found to be significant in our study, the changes observed are not necessarily consistent between the two studies. Both this study and Bleumel et al. observed a decrease in *Lactobacillus* and an increase in *Parabacteroides* and *Prevotella*. However, we observed an increase and larger population of *Bacteroides* in controls and a relatively stable *Bacteroides* population in ethanol treatments while Bleumel et al. reported an increase in ethanol treatments. Xu et al. identified *Helicobacter*, *Allobaculum*, *Turicibacter*, and *Adlercreutzia* as displaying significant changes in relative abundance [36]. While we do observe a slight decrease in *Helicobacter* relative abundance temporally in ethanol treatments (Figure 2 and Appendix A), we consistently see *Helicobacter* being dominant in the ethanol group vs. controls as a whole contrary to Xu et al.’s findings. Xu et al. reported an increase in *Turicibacter* where we saw a significant and consistent decrease to where abundance was negligible in the ethanol group (Appendix A). Both Xu et al. and our study found an increase in abundance and a targeted interest in *Adlercreutzia*.

Differences in findings on shift of significant genera between studies may be due to experimental design factors. For example, Bleumel et al. looked at ileum samples while we looked at fecal samples and our study tracked temporal dynamics while Xu et al. looked at a 3-week snapshot. This once again highlights the need for continued study across multiple independent experiments and studies in order to focus on treatment and temporal trends specific to ethanol treatment while minimizing individual experimental effects. The findings of our study across independent experiments along with commonalities in significant genera across independent studies, accounting for differences in diversity and abundance results, all point towards the existence of a common and characterizable “mouse ethanol gut microbiome” that is worth further exploration.

## 5. Conclusions

Our collective analysis highlights the importance of using multiple experiments and accounting for temporal factors in exploring ethanol-related changes to the mouse microbiome. While our results are consistent with many other studies in terms of taxa that are significant to the shift, differences in magnitudes and directions of shift exist. These differences are likely identifiable in future studies and may include factors such as colonization effects, variable starting microbiome communities, and differences in experimental design. The consistency observed in taxa identification and certain taxonomic and diversity trends across independent studies, despite variations in methods and approaches, together with phenotypic observations all point towards overarching and common effects of alcohol on the gut microbiome.

## Figures and Tables

**Figure 1 life-11-00007-f001:**
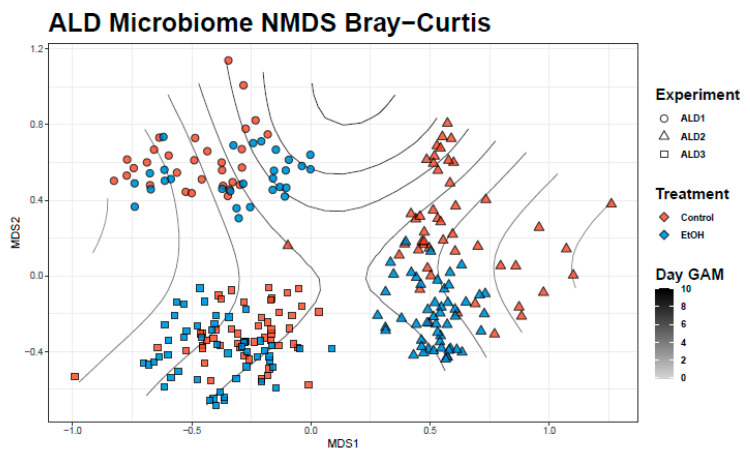
Non-metric multidimensional scaling (NMDS) plot of experiment and treatment samples using Bray–Curtis distance. Gradient lines represent temporal generalized additive model (GAM) fit to ordination space.

**Figure 2 life-11-00007-f002:**
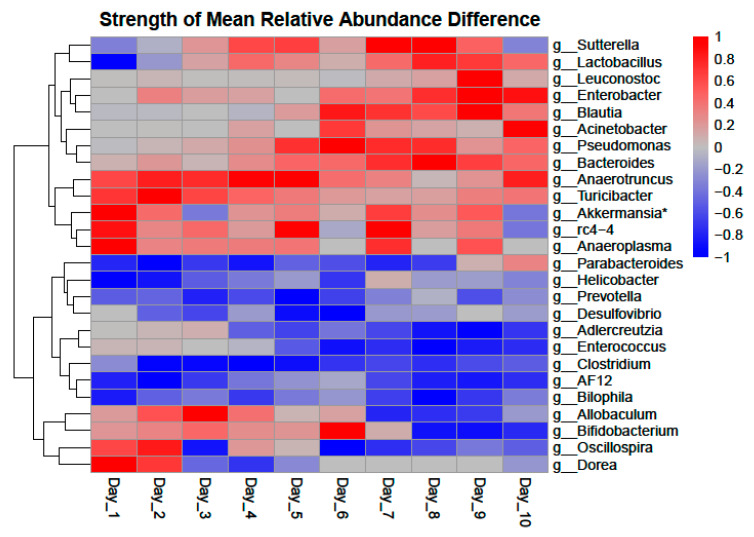
Strength of difference in mean relative abundance of Point-biserial coefficient analysis (PBCA) significant genera. A value of “1” or “−1” represents the maximum magnitude of difference seen in the genera over the time course with positive values (Red) favoring controls and negative values (Blue) favoring ethanol treatments. * *Akkermansia* was not significant in any analysis.

**Figure 3 life-11-00007-f003:**
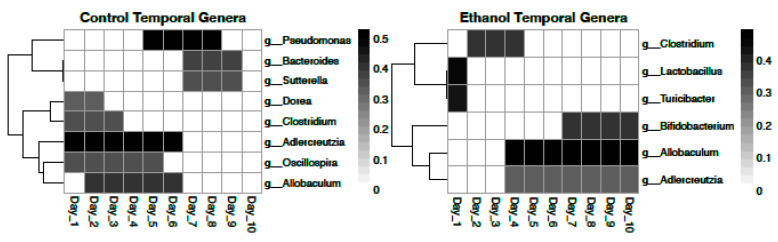
Heatmap representation of PBCA identified temporally significant bacterial genera in control and ethanol treatments. “Heat” is representative of *r_pb_* score for the range of days with darker colors representing a larger significant *r_pb_* score.

**Figure 4 life-11-00007-f004:**
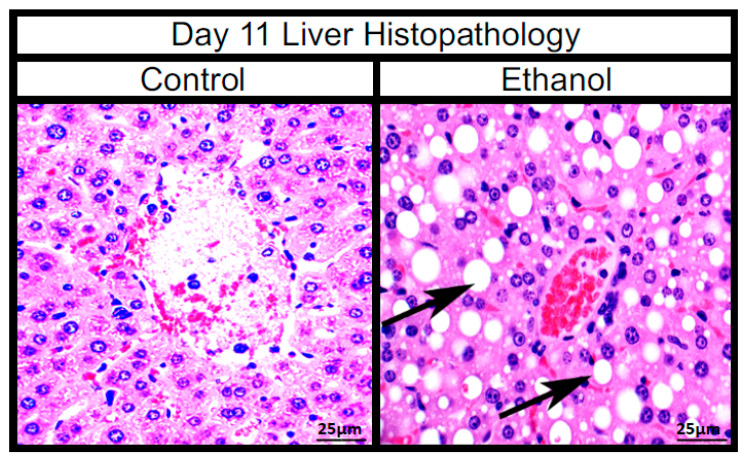
Images of day 11 mouse livers at 400× with H&E stain. Black arrows point to signs of macrovascular steatosis (fat deposits), visible as large white ovals in the ethanol treatment.

**Table 1 life-11-00007-t001:** PBCA identified significant bacterial genera for control and ethanol treatments. AF12 is a genus in the Rikenellaceae family and rc4-4 is a genus in the Peptococcaceae family.

Control	*r_pb_*	*p* Value	Ethanol RX	*r_pb_*	*p* Value
g__Pseudomonas	0.457	0.005	g__Clostridium	0.407	0.005
g__Bacteroides	0.426	0.005	g__Adlercreutzia	0.385	0.005
g__Anaerotruncus	0.263	0.005	g__AF12	0.293	0.005
g__Enterobacter	0.222	0.005	g__Bilophila	0.290	0.005
g__Turicibacter	0.206	0.005	g__Helicobacter	0.174	0.02
g__rc4-4	0.166	0.005	g__Prevotella	0.163	0.01
g__Anaeroplasma	0.152	0.03	g__Enterococcus	0.154	0.005
g__Blautia	0.150	0.005	g__Desulfovibrio	0.149	0.015
g__Acinetobacter	0.148	0.005	g__Parabacteroides	0.139	0.02
g__Sutterella	0.125	0.04			
g__Leuconostoc	0.096	0.005			

## Data Availability

Sequencing data that support the findings of this study have been submitted to the NCBI SRA under BioProject ID: PRJNA669074.

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
