# Peer review of "The Gut Microbiome and Alcoholic Liver Disease: Ethanol Consumption Drives Consistent and Reproducible Alteration in Gut Microbiota in Mice"

_life, 2020, doi:10.3390/life11010007_

Round 1

Reviewer 1 Report

Recent studies have suggested that ethanol-associated alterations in the gut microbiome contribute to the development of alcoholic liver disease (ALD).

Numerous human and animal studies have shown that alcohol affects the gut-liver axis at multiple interconnected levels, including alterations in the gut microbiome and decreases in intestinal epithelial barrier function leading to an increase in systemic exposure of gut bacteria and bacterial toxin culminating in the development of ALD.

In the present work, the authors look at microbiome shifts in a mouse model of ALD, consisting of chronic ethanol feeding plus binge (Gao-binge model) across three independent experiments.

Twenty-three mice were divided into two homogeneous groups (control and ethanol treatment) and were fed a nutritionally adequate liquid control diet (Lieber Di Carli Control Diet) for three consecutive days to establish a similar baseline gut microbiome. Fecal samples were collected during this acclimation period (day 0).

This phase was followed by feeding with Lieber Di Carli Diet containing 5% ethanol for ten days in the ethanol group mice while control mice were pair-fed with Lieber Di Carli diet without ethanol for the same ten days (days 1 - 10).

Three independent experiments named ALD1, ALD2, and ALD3 were performed at different times. ALD1 followed three mice, each in control and ethanol groups while ALD2 and ALD3 consisted of observing five mice in each group for a total of 13 mice each in control and ethanol groups across all experiments. Fecal samples were collected from individual mice cages each day and stored at -20ºC for DNA extraction. On the 11th day of ethanol feeding, mice in the ethanol group received gavage of a single dose of ethanol (5g/kg body weight), while mice in the control group received gavage of isocaloric dextrin maltose. After the 11th day, mice were sacrificed for liver histopathology.

Fecal samples from day 0 showed no significant difference in PERMANOVA or ANOSIM tests between control and ethanol groups.

Ethanol group bacterial communities and those of controls were significantly different from each other using both PERMANOVA (p<0.001) and ANOSIM (p<0.001) tests.

Differences between experimental groups and between experiments were visualized in NMDS ordination.

Across experiments, a significant decrease in mean Shannon (2=8.3, p=0.004) and Simpson (2=18.9, p<0.001) alpha diversity scores were observed for samples in the ethanol group. Weals observed a significant decrease in beta dispersion (p=0.02) in the ethanol group across the three experiments. PBCA identified significant ASVs (control = 150, ethanol = 165) which, was representative of approximately a third of total ASVs. PBCA identified 20 significant genera OTUs (control = 11, ethanol = 9).

The alcohol feed/binge caused macro vascular steatosis of hepatocytes and inflammatory infiltration consistent with alcoholic steatohepatitis in the ethanol group compared to controls by day 11 of the study.

The authors found significant differences in microbiome community assemblies between ethanol-treated mice and control mice. They also identified significant differences in gut microbiota occurring temporally with ethanol treatment. Peak shift was observed in the communities four days after the start of daily alcohol consumption. The authors quantitatively identified many of the bacterial genera indicative

of these ethanol-induced shifts including twenty significant genera when comparing ethanol treatments with controls and 14 significant genera based on the temporal investigation. Including overlap of treatment with temporal shifts, they identified 25 specific genera of interest in ethanol treatment microbiome shifts. Shifts coincide with the observed presentation of ALD associated phenotype.

The study is sophisticated but well described.

The figures and tables are clear and readable. The figure and table captions are complete and accurate.

Methods are well described. I suggest specifying that the overall population is composed of 23 mice.

Experiments are reproducible.

The study conforms to ethical guidelines of Pennsylvania State University.

The results support the conclusions. The discussion is good and deep.

For the statistical analysis, I do not have the expertise with R to consider the statistics.

There are some typos, so I recommend language editing.

For the reasons stated above, I recommend that you accept the paper after minor revision.

Author Response

We thank the reviewer for their detailed and thoughtful review, comments, and suggestions. The authors appreciate the reviewer's positive comments.

We find the reviewer's summarized conclusions of our findings accurate.

Specific reviewer comments and responses:

Reviewer: The figures and tables are clear and readable. The figure and table captions are complete and accurate.

Response: Thank you. We discovered that the captions for Figure 2 and Figure 3 to be reversed in the formatted review copy we were provided. We have corrected this in the revised submission.

Reviewer: Methods are well described. I suggest specifying that the overall population is composed of 23 mice.

Response: Thank you. We have re-written lines 75 to 77 to better express total mice in the study in the revised submission. The new text states "A total of 26 mice (13 control and 13 ethanol-treated) were included in this study. ALD1 consisted of 6 mice (3 control and 3 ethanol-treated) while ALD2 and ALD3 each consisted of 10 mice (5 control and 5 ethanol-treated).

Reviewer: There are some typos, so I recommend language editing.

Response: Thank you. We have additionally proofread the document and made the following related corrections

  • Line 21 removed erroneous whitespace.
  • Line 39 "and" to "an".
  • Line 49 "on" to "of".
  • Line 99 "16s" to "16S".
  • Line 112 "Organizational" to "Operational".
  • Line 120 "Generalized Additive Models" to "generalized additive models".
  • Line 126 "Ethanol" to "ethanol".
  • Line 129 added ",".
  • Line 132 "a priori" italicized.
  • Line 136 "16S" in header corrected.
  • Line 137 "16s" to "16S".
  • Line 141 "Day" to "day".
  • Line 143 removed erroneous whitespace.
  • Line 177 corrected to reference Figure 3 instead of Figure 2.
  • Line 177 erroneous whitespace removed.
  • Line 251 "gram positive" to "gram-positive".
  • Line 262 "sontrols" to "controls".
  • Line 268-270,275,287,296 changed 7 instances of "Gram" to "gram".
  • Line 298 "Controls" to "controls" an "Ethanol" to "ethanol".
  • Line 314 removed erroneous "as".
  • Line 359 added missing whitespace character.
  • Line 363 capitalized grant number character
  • Figure 2 caption, line 204 "Controls" to "controls" and "Ethanol" to "ethanol"
  • Table 1 caption, line 303 "Controls" to "controls" and "Ethanol" to "ethanol".

Reviewer 2 Report

Could the authors please clarify how the numbers of animals used in their experiments were determined i.e. how were their experiments powered?

Author Response

We thank the reviewer for the time taken to review our manuscript and their comments.

Specific reviewer comments and responses:

Reviewer: Could the author's please clarify how the numbers of animals used in their experiments were determined i.e. how their experiments were powered.

Response: Thank you for the question. ALD1 was a preliminary experiment where space and resources dictated that 6 mice (n=3) would be an adequate choice. ALD2 and ALD3 were limited to 10 mice each (n=5 in each) by space and cage availability. This contributed to the decision to integrate several experiments in order to increase power in the study. We have additionally re-written lines 75 to 77 to better express total mice in the study in the revised submission. The new text states "A total of 26 mice (13 control and 13 ethanol-treated) were included in this study. ALD1 consisted of 6 mice (3 control and 3 ethanol-treated) while ALD2 and ALD3 each consisted of 10 mice (5 control and 5 ethanol-treated).

Reviewer 3 Report

This is an original article regarding gut microbiome and alcoholic liver disease. 

The topic is promising but is not new in the scientific literature.

In this context, Please consider the following articles (to be discussed and included)

Scarpellini E, Forlino M, Lupo M, Rasetti C, Fava G, Abenavoli L, De Santis A. Gut Microbiota and Alcoholic Liver Disease. Rev Recent Clin Trials. 2016;11(3):213-9. doi: 10.2174/1574887111666160810100538

Wiernsperger N. Treatment strategies for fatty liver diseases. Rev Recent Clin Trials. 2014;9(3):185-94

It would be interesting to highlight the clinical implications of this research

The results of the study are well described and very interesting (the analyzes are also complete)

The discussion must be shortened and focused on the main topic. 

The introduction can also be improved by highlighting the aim of the study

Author Response

We thank the reviewer for their thorough, detailed review and thoughtful comments.

Specific reviewer comments and responses:

Reviewer: The topic is promising but is not new in the scientific literature.

Response: Thank you for being candid. The goal of this manuscript is not necessarily to open a new topic of study but rather to further research in the field and offer new insights into microbiome dynamics in support of previous work while expanding and improving understanding for ongoing and future research.

Reviewer: In this context, Please consider the following articles (to be discussed and included)

Response: Thank you for sharing these very interesting articles. We have incorporated them as references at relevant locations in the paper (Wiernsperger for treatment difficulty in the Introduction)(Scarpellini et al. for gut-liver access in the Introduction). In the interest of concerns about the length of discussion and desire for additional clinical implications addressed below, we did not focus specifically on these papers with additional manuscript content at this time. Those discussion will certainly be quite relevant in more clinical focused work we that we have ongoing.

Reviewer: It would be interesting to highlight the clinical implications of this research

Responses: We acknowledge that this manuscript is weighted more towards microbial ecology than clinical implications in its presentation. We have ongoing and future work of a much more clinical nature related to this work that we are eager to conclude and share.

Reviewer: The results of the study are well described and very interesting (the analyzes are also complete)

Response: Thank you. This comment is appreciated by the authors.

Reviewer: The discussion must be shortened and focused on the main topic.

Response: We acknowledge that the discussion is somewhat long. However, we feel the included topics and details are necessary given other (sometimes conflicting) reported findings in prior work, and are appropriate to our findings. Some reviewers acknowledged the depth of the discussion as positive, and after additional review by the authors, we struggle to find obvious sections to cut.

Reviewer: The introduction can also be improved by highlighting the aim of the study

Response: Thank you for this comment, we believe this has helped us shape the introduction in a positive manner. We have added context to the final paragraph of the introduction (line 62-64) to help more clearly define goals. Statement with new text: "Here we look at gut microbiome shifts in a mouse model of ALD, consisting of chronic ethanol feeding plus binge (Gao-binge model) across three independent experiments, in order to explore treatment-based and temporal effects related to alcohol consumption and ALD."